# Recent Advances in Coupled MBS and FEM Models of the Spine—A Review

**DOI:** 10.3390/bioengineering10030315

**Published:** 2023-03-01

**Authors:** Kati Nispel, Tanja Lerchl, Veit Senner, Jan S. Kirschke

**Affiliations:** 1Associate Professorship of Sport Equipment and Sport Materials, School of Engineering and Design, Technical University of Munich, 85748 Garching, Germany; 2Department of Diagnostic and Interventional Neuroradiology, School of Medicine, Klinikum Rechts Der Isar, Technical University of Munich, 81675 Munich, Germany

**Keywords:** multibody simulation, finite element method, co-simulation, spine, spinal loading, sports, biomechanics, degeneration, intervertebral disc, coupled

## Abstract

How back pain is related to intervertebral disc degeneration, spinal loading or sports-related overuse remains an unanswered question of biomechanics. Coupled MBS and FEM simulations can provide a holistic view of the spine by considering both the overall kinematics and kinetics of the spine and the inner stress distribution of flexible components. We reviewed studies that included MBS and FEM co-simulations of the spine. Thereby, we classified the studies into unidirectional and bidirectional co-simulation, according to their data exchange methods. Several studies have demonstrated that using unidirectional co-simulation models provides useful insights into spinal biomechanics, although synchronizing the two distinct models remains a key challenge, often requiring extensive manual intervention. The use of a bidirectional co-simulation features an iterative, automated process with a constant data exchange between integrated subsystems. It reduces manual corrections of vertebra positions or reaction forces and enables detailed modeling of dynamic load cases. Bidirectional co-simulations are thus a promising new research approach for improved spine modeling, as a main challenge in spinal biomechanics is the nonlinear deformation of the intervertebral discs. Future studies will likely include the automated implementation of patient-specific bidirectional co-simulation models using hyper- or poroelastic intervertebral disc FEM models and muscle forces examined by an optimization algorithm in MBS. Applications range from clinical diagnosis to biomechanical analysis of overload situations in sports and injury prediction.

## 1. Introduction

When humans evolved to adopt an upright position, the spine became a central structure of the human biomechanical system. As such, it can be a source of pain that can significantly impact a person’s quality of life. Intervertebral disc (IVD) degeneration and herniation are possible causes of back pain. Research has shown that both aging [1] and overload [2,3], often experienced by ambitious athletes, can contribute to the degeneration process. However, the relationship between abnormal loading and degeneration is not fully understood. In addition, IVD changes that are visible in magnetic resonance imaging (MRI) have not proven to be clear evidence of back pain [4]. To better understand the mechanisms underlying back pain and identify potential solutions, it is important to study the spine and its related structures in a holistic manner, considering the interactions of motion or posture, pain, and IVD biomechanics. However, the difficulty of directly observing these structures in vivo makes this task challenging. Numerical simulations can be useful in modeling and analyzing the mechanics of the spine, to identify factors that contribute to spinal health problems and potential interventions.

Multibody simulation (MBS) is widely used to gain insights into the healthy and pathological biomechanics of the spine from a macroscopic perspective. MBS models study joint reaction forces, muscle forces and muscle activation patterns in combination with respective movements or positions. There are several such models that include intervertebral joints with three rotational [5,6,7,8,9,10,11,12,13,14,15,16,17,18,19] or six translational [20,21,22] degrees of freedom (DoFs) while neglecting passive structures, such as ligaments and facet joints. Using force elements as additions to joints allows individual stiffness definitions for all given DoFs [15,16,17,18,19,20]. For all models, including rotational joints in at least one dimension, the position of joints is a central issue, as it defines the centers of rotation and therefore, directly influences joint kinematics. The fixed centers of rotation in these studies were set either in the center of the IVD [6,12,17], or according to Pearcy and Bogduk [23], in the posterior half of the upper endplate of the inferior vertebra of each motion segment [9,15,16,24].

Finite element method (FEM) spine models aim to simulate deforming bodies in a detailed manner to reveal their inner stresses. Models may include the whole spine, a segment or only an IVD. Most models contain manually created geometries for IVDs and vertebrae [25], but recent approaches use automated segmentation algorithms to derive patient-specific geometries [26,27]. For IVDs, the current gold standard is a biphasic, poroelastic and nonlinear model including an anulus fibrosus (AF) and a nucleus pulposus (NP) component. While bone is usually modeled with linear material properties [28], or as a rigid body [29], material properties for the IVD commonly include hyperelastic material models, such as the Neo-Hooke and Mooney–Rivlin models [30]. As IVD degeneration is believed to affect IVD biomechanics [31,32], several approaches also deal with degeneration-dependent material properties [33,34]. A key evaluation criterion of FEM spine models is their ability to describe arbitrary material behavior of the IVD while numerically reporting all its biomechanical functions, such as load transfer and bulging events [35].

In summary, MBS models are suitable for investigations of the overall kinematics and kinetics of the trunk, or even the full body. However, detailed information on flexible body deformation and internal stress cannot be provided. Complex, heterogeneous structures, such as the IVD, or even entire functional spinal units (FSU) combining all stabilizing structures (IVD, ligaments, facet joints), are reduced to a resultant mechanical response to external deformation, often in reference to an approximated center of rotation. In contrast, FEM models are beneficial for simulating detailed structures and analyzing inner stresses and deformations. However, detailed FEM meshes may cause large computational costs, and thus, models often include simplifications. Additionally, defining realistic boundary conditions (BC) for FEM models remains a main challenge due to the lack of possibilities for in vivo measurements. Until now, no satisfactory, broad investigations are available containing this data [22].

Coupling MBS and FEM takes advantage of the strengths of each method and offers great potential to analyze the deformations and stresses of IVDs while considering the overall kinematics and kinetics of the trunk. How coupling is implemented can be subdivided into two approaches: One is to calculate acting forces or present deformations completely before starting the other simulation (unidirectional data flow). The second approach considers acting forces or displacements anew in every increment of an ongoing simulation, generating a constant, bidirectional data flow. Unidirectional coupling commonly includes an inverse dynamic MBS simulation with resulting forces and moments, which are subsequently used as loading and BCs in an FEM model [36,37,38,39,40,41,42,43]. Less common approaches use FEM model displacements and stresses to define properties of force elements, occasionally called bushing components, in MBS simulations [44]. Few approaches apply order-reduction techniques to the FEM model before integrating it into an MBS model [45], which reduces computational costs in linear models, but does not support parameter variation after the reduction is carried out. In general, unidirectional coupling is suitable for linearly deforming FEM bodies because the positions of the body’s particles do not change substantially, providing the possibility of directly applying the calculated forces of the MBS simulation subsequently to the respective locations in the FEM bodies. Bidirectional coupling is preferential whenever large deformations are present [46,47]. In that case, the deformations of the flexible bodies have a direct impact on multibody kinematics, hindering the option of consecutive simulations.

Hence, by using co-simulation, we can examine the pathological spine in a holistic way to analyze the interactions of various factors, such as motion, posture, back pain and the biomechanics of the IVD. In this work, we review advances in coupled MBS and FEM simulations of the spine. Specifically, we excluded reduced order models and instead focused on the type of data exchange, whether unidirectional or bidirectional, as this is essential for the simulation of large deformations and constant interactions of different components in the spine.

## 2. Methods

Coupling MBS and FEM models has become a common practice in spine modeling during the last decade. However, no terminology has been established yet to differentiate between what are here called unidirectional and bidirectional co-simulation. To achieve comprehensive coverage of studies using co-simulation, the following keywords were searched in several combinations in different databases: coupled, spine, multibody, MBS, FEM, hybrid, finite element, co-simulation, combined. Resulting articles were manually reviewed and included only if they: (i) contained a type of coupling between MBS and FEM and (ii) were published after 2009 or included models that were still used in studies published after 2009.

## 3. Unidirectional Co-Simulation of the Spine

We first reviewed MBS and FEM models of the spine using unidirectional coupling, which was characterized by the execution of one simulation and the integration of the results into another simulation. Considering the order of execution, three distinct methods of coupling were identified: MBS execution and transfer of data to an FEM simulation; FEM and transfer of data to MBS; and a threefold data transfer from FEM to MBS and to FEM again. Figure 1 visualizes the representative simulation models with the different coupling methods.

### 3.1. MBS → FEM

A unidirectional data transfer from MBS to FEM was the most common approach in unidirectional coupling. Muscle forces, reaction forces or displacements were calculated by an MBS simulation and integrated as boundary conditions (BC) in a subsequent FEM simulation.

Esat et al. [37,38] established a unidirectional MBS and FEM co-simulation of the cervical spine to calculate intradiscal pressure and AF stresses at large impact accelerations. Geometries for rigid MBS bodies were inspired by human anatomy but not specifically derived from medical images. Bodies were created in computer-aided design (CAD) software and imported in Nastran (MSC Software, Garching, Germany) before ligaments and facet joint contacts were integrated. IVD joints were modeled with nonlinear viscoelastic bushing components. One hundred and forty muscles with their respective time-dependent forces were transcribed using a dynamic calculation framework (Virtual Muscle 3.1.5 [48]). The FEM model was then implemented with independent dimensions and locations of IVDs, which were derived from the general quantitative anatomy of the spine in the software Marc/Mentat (MSC Software, Garching, Germany). NP and AF were defined using linear material parameters. Two impact loadings, namely, 15 g frontal and 8.5 g rear-end impacts, were implemented at the neck of the MBS model. The resulting loads—two sagittal forces and one sagittal moment—at the interface points for each IVD, were measured in the MBS model and implemented as time-dependent force BCs in the subsequently set-up FEM model to predict von Mises stresses in the AF and the intradiscal pressure in the NP.

Du et al. [49] investigated the impact of ejection out of, e.g., a plane on the IVDs using undirectional co-simulation. The MBS model included the entire human body seated in a chair and consisted of 16 rigid elements, with the spine divided only into the neck, upper body and lower body. Joint properties were taken from the literature. The joint between the upper and lower torso was replaced by a spring with a stiffness level of 900 N/mm. Loading was applied through an accelerative load peak of 15 g. The nonlinear FEM model of the thoracolumbar–pelvis complex (T9-S1) included the following material properties: vertebrae and pelvis as isotropic homogeneous elastic material, IVD divided into NP and AF, with the NP containing brick elements of an incompressible, hyperelastic Mooney–Rivlin material model and the AF containing hyperelastic ground substance and 3D cable elements as fibers with nonlinear stress–strain behavior, along with seven ligaments as tension-only cable elements. Attachment points and cross-sectional areas of the ligaments were obtained from Agur et al. [50]. The model was validated statically and quasi-dynamically. The MBS model was used to calculate the translation and rotation of freely chosen reference points (RP), namely, the hip joint and a point at the center of the superior endplate at T9, which were subsequently implemented as time-dependent BCs in the FEM model. For both models, Hypermesh (Altair Engineering Corp., Troy, MI, USA) was used.

Henao et al. [51] simulated surgery procedures to predict potential spinal cord damage. To do so, they implemented a complete, patient-specific FEM spine model that included the spinal cord. A previously developed MBS model of the spine [52] contained 6-DoF springs as IVD and ligament representatives. It was used to calculate the displacements of respective vertebrae, which were then incorporated as BC in the FEM model. No information was given on the stiffness parameters of the spine and how the authors ensured consistency with the FEM IVD model. The study was rated useful for surgery planning, as spinal cord injuries could be predicted by the model. Individualized parameters, especially in the area of the IVD, would likely improve prediction quality.

Honegger et al. [53] investigated the IVD stresses during the sit-to-stand (STS) transfer of lower-limb amputees using a unidirectional co-simulation. The lower-body MBS model was built in OpenSim and included 294 Hill-type muscles and five lumbar vertebrae, which were complemented with three DoF bushing elements containing stiffness values found in the literature [20]. Simulating the STS motion in the MBS model resulted in the lumbar pelvic rhythm, namely, the respective time-dependent joint angles of the lumbar spine and the muscle forces. In a lumbar FEM model taken from Campbell et al. [54], these values were introduced as time-dependent inputs, together with joint contact forces and moments. Results included, among others, the time-dependent AF stresses, facet joint loads and IDPs during the STS. Comparing the IDPs calculated based on the MBS joint force and the IDPs found in the FEM model showed greater agreement between MBS values and in vivo data [55]. The authors stated that personalized data would possibly improve the muscle force and joint-load estimations from the MBS model, and thus have an impact on the FEM model and final simulation results.

A research group bulit around Shirazi-Adl and Arjmand has been developing numerical models of the spine since 1985 [56] and has started to advance, in parallel, two distinct model versions, which have been known by various names throughout the years. For overview reasons, the two models are herein marked as “I” and “II”.
(I)The first model included an initial, detailed FEM representation of the spine.(II)The second model was based on the detailed model but consisted of rigid bodies and interconnecting beam elements. It is later referred to as the musculoskeletal (MS) model.

Version II was not solved with a classic MBS solver, but included clear characteristics of an MBS model, such as the involvement of mainly rigid bodies and their interconnections by joint-like components. While version II could not represent detailed deformations, it was able to perform large numbers of nonlinear analyses [57]. The researchers refined these models into a passive, osteoligamentous (I) and an active, muscular part (II) and considered their respective roles in a coupled, biomechanical system [57,58,59]. They implemented their passive FEM model in Abaqus (Dassault Systèmes, Paris, France), consisting of single rigid vertebrae (L1–L5) and one rigid body combining T1–T12. The IVDs were created by extruding the endplate geometries and were infused with rebar elements as AF fibers. The active model was based on a Fortran code. For the coupling process, virtual springs were attached to each vertebra in the passive model (I) to allow the transmission of shear forces and axial, lateral and sagittal moments. Hence, the load sharing between the passive structures and the muscles could be controlled.

In 2002, Shirazi-Adl et al. [60] incorporated a novel, kinematic-based Matlab (the Mathorks Inc., Natick, Massachusetts, USA) algorithm into the simplified model (II) to interactively calculate optimized solutions for muscle- and passive reaction forces. Refer to [61] for a detailed data flow chart. Note, optimization algorithms are commonly used in MBS models for muscle force estimations as well. Beam joints in the active model (II) represented the overall nonlinear stiffness levels of the spinal segments—namely, IVDs, facets and ligaments—with a nonlinear load–displacement curve, which was defined in an iterative process towards yielding the best agreement with the detailed FEM results (I) [57] and was direction-dependent [60]. Ten muscles, five global and five local, were included with six distinct fascicles [62]. The detailed FEM model (I) was based on CT images and consisted of 6 vertebrae (L1-S1) as rigid bodies, five IVDs divided ino AF and NP, 10 contact facet surfaces and 9 sets of ligaments [40]. Fourteen AF lamellae were considered with rebar fibers and a linear elastic ground substance, along with an incompressible NP. Satisfactory agreement considering predicted rotations under flexion, extension and torque loading validated the two respective models against each other [60]. Given IVD cross sectional areas, values in the FEM model (I) were 17% smaller than the ones in the MS model (II) [36].

Azari et al. [36] modified the models by applying the idea of a follower load (FL), which is a method for including simplified muscle forces in passive FEM models (I) to achieve more realistic load conditions [40]. The FL’s line of action was aligned with the lumbar spine’s curvature, passing the endplates and vertebrae at the approximate center. The FEM model was based on the detailed, passive FEM model (I), but included only the L4–L5 segment. The MS model (II) was adopted mainly unchanged from earlier work by the research group [56,57,60,61,62]. To estimate stresses and strains in IVDs; ligaments and facets; and load sharing among these structures under realistic loading conditions, Azari et al. applied gravity loads and muscle forces from an MS trunk model (II) to the passive FEM model (I) in several static positions. Twelve static positions were simulated based on the availability of the results of Wilke et al. [55] and the kinematic data gathered by Arjmand et al. [62]. A resultant force including all gravity loads and muscle forces with upper insertions at and above L4 was calculated using the MS muscle force predictions. After the passive FEM model was rotated manually per IVD to fit the kinematic position of the twelve respective MS models, the research group tried two approaches: directly applying muscle forces estimated by the MS model in the center of the L4 body, and determining a substituting FL by trial and error to yield a similar IDP as derived by the previous approach. The FL was applied as a unilateral, pre-compressed spring between the L4 and L5 centroid.

To analyze scoliotic spine loads and their growth patterns, Kamal et al. [39] established a combined MS and FEM simulation based on subject-specific upright positional data. An MS model data of the bony spine components, including the pelvis, hip and ribcage, was derived from subject-specific CT images using mimics and was subsequently aligned into an estimated upright position with the aid of optical imaging tools. IVD centroids were considered the centers of rotation (CoRs) for the modeled spine section T12–L1 and set up with three nonlinear rotational stiffness values according to Hajihosseinali et al. [17]. Simulations were run statically with all rigid bodies fully constrained. Muscle attachment points for more than 160 muscles were calculated using Matlab. A static optimization algorithm then computed muscle forces and reaction moments, namely, individual forces F*_m_* and reaction moment M*_CoR_*, for each muscle. Reaction forces were considered as the sum of gravitational forces and muscle forces. The resulting forces and moments in a static equilibrium condition calculated in the MS model were applied to the vertebral growth plate FEM bodies as distributed pressures and shear stresses. Note, IVDs were solely considered as joints, and the biomechanical focus lay on the growth plates. FEM simulations were carried out with Abaqus (Dassault Systèmes, Paris, France). The resulting stresses in the growth plates were comparable with results from the literature and may be helpful in predicting growth patterns in scoliotic spines.

Further studies included a hybrid model applying muscle forces calculated in the beam-joint-based MS model (II) to the updated, passive T12-S1 FEM model (I) [40]. The MS model consisted of 56 muscles and was solved with an optimization algorithm minimizing the sum of the cubed muscle stresses. The outcome served as a basis for evaluating the loads on the beam joints. Simulations included eight static tasks for which the muscle forces were estimated by connector elements between muscle insertion points in the MS model. Together with the pelvic rotations gathered from in vivo experiments, these forces were substituted into the FEM model. Final deformed positions for the eight tasks were found to vary between the MS and FEM model, and the authors subsequently applied an iterative process, in which the compression-dependent stiffness properties of the MS beam-joints were adapted and resulting forces were again prescribed into the FEM model until the solution was convergent. The hybrid nature of the approach was thus the adaption of the beam-joint parameters based on a passive FEM model deformed position, which was defined actively by MS results. Kinematics of the MS and FEM models differed by less than 1 mm in the final state.

In order to develop a self-defined gold standard of spine modeling, Rajaee et al. [63] further developed the hybrid MS and FEM model introduced by Khoddam-Khorasani et al. [40] towards a coupled model, which enhances the hybrid model by incorporating the musculature of the MS model (II) in the FEM model (I). The manual approach of iteratively updating the nonlinear beam-joint stiffness was thus avoided. The resulting model hence consists of the rigid vertebrae defined earlier in the passive FEM model (I), the detailed geometries of the IVDs and all muscles and ligaments defined in the MS model. In the resulting FEM model, muscle forces are predicted by a procedure similar to the optimization procedure described above [60], and due to the novel method, depend on detailed IVD FEM model deformations. Static positions were simulated to evaluate the performance of the model compared to earlier versions and in vivo data. Stress distributions in IVDs were not available in the study, but compression forces and muscles forces could be compared between different model versions.

To sum up, multiple simulation approaches have been executed using MBS simulation results in FEM models. Transferred data included reaction forces, muscle forces or displacements of vertebrae on different spine levels. The vast majority of MBS studies rely on experimental data to define joint properties of the IVDs. However, one study investigated the potential of FEM simulations to define MBS model properties.

### 3.2. FEM → MBS

Karajan et al. [44] investigated the possibility of representing IVD behavior in MBS bushing components by polynomial functions previously derived from detailed FEM simulations. In their approach, one FSU was simplified with three cylinders. The IVD model was constructed based on earlier work combining porous and polyconvex material to account for the complex behavior of the IVD [35]. The conversion of the FEM displacements into the three-DoF MBS bushing-element parameters, namely, rotation and translation in the sagittal plane, was realized as follows: From the center of gravity in the IVD, where the bushing component was located, lever arms to the nodes on the top surface of the IVD aid in yielding nodal displacements of surface nodes. Nodal deformations were then summarized in a Cardan rotation tensor. The FEM simulation with the nodal displacements as input yielded a surface-traction vector derived from the reaction stresses of the IVD, which was homogenized by summarizing the stress distribution to one single value for the force and moment acting on the surface. With the resulting force and moment, the load–displacement behavior of the IVD at the center of gravity was computed. However, as their work focused on the coupling scheme, IVD responses were investigated solely for the elastic part of an MBS bushing component, and nonlinear deformations were not considered. Karajan et al. stated that the advantage of the method was the significantly shorter computation times compared to other approaches. The co-simulation aspect here consists of the modification of the bushing component definitions in the MBS analysis based on the FEM simulation results. The authors stated that the developed model yields the same elastic response of the IVD as that expected by the full biphasic FEM model in each time step.

The resulting MBS model with adapted stiffness parameters could potentially be further used to provide information for a detailed FEM simulation, calculating stresses and deformations. This step has been realized by some research groups. Studies are reviewed in the following.

### 3.3. FEM → MBS → FEM

To investigate the effects of muscle damage as seen in lumbar interbody fusion surgery, Kumaran et al. [64] used multiple data exchanges between an OpenSim MBS and an Abaqus FEM model. Firstly, an existing FEM model of the thoracolumbar spine [65] was loaded with 4 Nm at T1 to gain ranges of motion (ROMs). These were then transferred to an OpenSim MBS model [8] to calculate muscle forces, which were subsequently applied to the lumbar part of the previously mentioned FEM spine model as connector forces. The research group noted that a limiting factor of their model was that these muscle forces did not produce the motion of the vertebrae, but the simulation was implemented with a given displacement to synchronize both models. Thus, the correct interaction of forces and displacement was likely not given.

In 2021, Meszaros et al., presented a study in which they adapted an established neuro-musculoskeletal spine model [22] to match with an FEM model of the spine [66] considering mechanical behavior. They used the Visible Human Male (VHM) of the Visible Human Project (VHP) [67] as an MBS basis model. From the patient-specific FEM model, they derived muscle attachment points (a) and patient-specific bone geometries (b), which were then subsequently morphed into the VHM MBS framework. The MBS framework was further individualized by generic structures (c), namely, joints, muscles, ligaments and IVDs, which were firstly customized in the neuro-musculoskeletal model and then prescribed to it. Next, soft-tissue characteristics of IVDs and ligaments within the VHM framework were adapted based on IVD responses in the FEM model, ligament stiffnesses derived from the FEM model and soft-tissue models in the neuro-musculoskeletal model. Scaling and simulating the resulting MBS VHM model led to time-dependent forces of muscles, tendons and ligaments ($F_{MTU(t)}$). These forces were finally inserted into the FEM VHM model together with the adapted bone geometries (b). The result of Meszaros et al.’s work was therefore an Abaqus FEM model with time-dependent forces $F_{MTU(t)}$ derived from an MBS model. The model allowed for investigations of spinal motion and tissue mechanics on a mechanical level.

Load sharing in the lumbosacral spine was explored by Liu et al. [42] in 2018 using a unidirectional MBS and FEM co-simulation. The MBS model was set up in Anybody (AnyBody Technology A/S, Aalborg, Denmark) with three DoF IVD joints at the centers of the instantaneous axes of rotation, respectively. Stiffness curves were predicted previously by FEM models of FSUs—they were devoid of ligaments and facet joints and loaded with flexion and extension moments. The model included 188 muscles of three different types: straight, via-point and nonlinear. Seven ligaments with fourth-order polynomial force-deformation relationships with respect to the spinal level [68] were added. The FEM model was implemented in Hypermesh and Abaqus based on the MBS model geometry. Endplates were meshed with shell elements, extruded to form brick elements as far as the adjacent vertebrae and divided into NP and AF. While bones and endplates were modeled with linear material parameters, hyperelastic Mooney–Rivlin models were used for the NP and AF ground substance, and nonlinear AF fiber parameters included increasing stiffness towards the outer lamellae [56,69]. Ligament locations and parameters were copied exactly from the MBS model. Moreover, the model included frictionless facet joint contacts. A static equilibrium was calculated in the MBS model following the concept of a spinal rhythm, in which the single FSUs were flexed proportionally and loaded by gravity. The reaction moment, ligament forces and muscle forces were applied to the T12–L1 joint of the FEM model. Results showed a different deformed state of the FEM model afterwards, which Liu et al. explained with the inability of the joint models in the MBS to allow deformations, which was in turn represented in the FEM model. To synchronize the models, Liu et al. translated vertebra L1 along the reaction force F*_R_* line of action until it reached to MBS-predicted position. The novel reaction force R*_F_* obtained was iteratively compared with the initial reaction force F*_R_* while adjusting the translation, until the difference was smaller than a predefined tolerance. In a follow-up work, Liu and El-Rich used the same model, reduced to one functional spinal unit (L4–L5), to investigate the influences of the NP position on the IDP, spinal loads and load sharing during 60° forward flexion. Based on in vivo data, three posterior shifts of the NP were realized by the models: 0, 1.5 and 2.7 mm. Muscles and ligaments forces, and joint forces and moments at L3–L4 were calculated by the MBS model and prescribed to the FEM model. IDP and spinal loads calculated by the FEM model show that the IDP and compressive forces within an FSU were distinctly influenced by the posterior shifts of the NP, and the CoRs calculated by their MBS and FEM model differ. Liu and El-Rich believe the kinetic results predicted by the MBS model to have been affected by single IVD rotating joints and suggest implementing an iterative process combining MBS and FEM models to account for compressive and shear stiffness. [42]

Refer to Table 1 for an overview of the reviewed unidirectional co-simulation studies. Author groups with more than one study mentioned in this review were included in the table only with their most recent studies for clarity reasons.

Independent of the data transfer being solely from MBS to FEM, from FEM to MBS or both, unidirectional co-simulation was often limited by linear deformations and manual adaption processes to synchronize both models [36,40,42]. To overcome these limitations in the field, a few recent studies implemented bidirectional co-simulations of the spine.

## 4. Bidirectional Co-Simulation of the Spine

Bidirectional co-simulation models benefit from an iterative data exchange that requires less manual intervention and more accurately accounts for large deformations of, e.g., IVDs. Approaches using this type of coupling are listed below, again focusing on how data are exchanged between MBS and FEM models.

Monteiro et al. [46] realized a bidirectional data flow to explore intersomatic fusion biomechanics. In their study, the MBS framework included specific reference points, for which the kinematic data, namely, displacements and rotations, were calculated. The results were transferred as initial data to the FEM simulation, which calculated the reaction forces and moments. Those were transmitted back to the MBS software and served as new starting points for another forward dynamic iteration. The process was managed by a co-simulation module, which used a gluing algorithm (an algorithm for bidirectional coupling of numerical models) to communicate between the MBS and FEM model of a C5-C6 or C6-C7 segment. This gluing algorithm was an adaptation of an algorithm developed in 2001 by Tseng et. al. [70], which in turn was based on the coordinate split (CS) technique by Yen et al. [71]. Wang et al. [72] complemented the gluing algorithm with an interfacing communication platform to make it suitable for practical applications. This approach made the single submodels black boxes, which could be coupled with one of three distinct algorithms distinguished by the data provided by the MBS system. Monteiro et al. applied the algorithm in which the MBS system (coordinator) provided kinematic data, as it was more convenient in the case of a forward dynamic analysis with given displacements. The general environment was developed in Abaqus and Apollo, a multibody system dynamics (MSD) simulator based on the Adams–Moulton method. The MBS algorithm was implemented in Fortran code. The co-simulation module core was incorporated into the Apollo code, and the co-simulation partner was Abaqus. The geometry of the IVD was derived from images and divided into NP and AF based on ratios from the literature. Material parameters were chosen as viscoelastic and quasi-incompressible with a hydrostatic NP and a fiber-drawn AF [37]. IVDs non-adjacent to the fused vertebrae were modeled as bushing components for efficacy reasons. Seven ligaments were introduced as viscoelastic, nonlinear elements. For the linkage of the MBS and FEM system, Monteiro et al. implemented so-called co-simulation elements as RPs. The possibility of more than one RP per linkage was mentioned as an option to consider more complex deformations. In his work, however, one master RP and one slave RP were introduced, either to the center of the top side of the IVD (slave) or to the center of the bottom side of the IVD (master). Refer to Figure 2a for a graphical representation of these RPs. The master RP belonged to the constrained master body, while the slave RP drove the deformation of the model. The information flux therefore consisted of the three basic stations: First, the kinematic data of the RPs were analyzed and stored. Second, the FEA was launched by taking into account the kinematic data of the RPs. Third, the kinetic results of step two were processed by the coordinator software. The validation of the model with a sagittal moment of 1.5 Nm applied to the head showed realistic results, confirming the compatibility of the MBS software with the FEM analysis considering the modeling of the spine.

Another coupling method was implemented by Dicko et al. [73] in combination with a composite lumbar spine model. The algorithm worked by dividing the lumbar spine model into particles with either rigid- or flexible-body characteristics. The vertebrae were represented by rigid bodies, each comprising a rigid coordinate system. The IVDs were represented as a discretized, meshed FEM body containing both rigid and flexible particles: Endplate particles were modeled as rigid-body particles; thus, they remained the same distance from each other over time. The remaining part of the IVD consisted of flexible particles, each experiencing an independent deformation (Figure 2b). The advantage of this approach was the reduction of unknowns, as FEM nodes could be attached to rigid-body particles using Lagrange multipliers. The movement of these attached nodes was then fully prescribed by the rigid body movement of the vertebrae, resulting in a reduction in the size of the equation system. A multimapping step reunited all particles before FEM parameters such as inertia and material properties were applied. The authors state that the model delivers accurate results without penalizing precision. The method was inspired by Stavness et al., who already implemented it in their software, ArtiSynth (Vancouver, BC, Canada) [74]. The software defines deformable bodies by representing their nodes as three DoF particles. Together with the other type of dynamic components—six-DoF rigid bodies—the model can be formulated as an ordinary differential Equation (ODE) and solved by a semi-implicit integrator. While particle-based approaches such as ArtiSynth are particularly well-suited for coupled biomechanical simulations, FEM models are approximated by a lumped mass matrix and a linear co-rotation, and other rotation effects are neglected. Linear FEM representations might not be able to adequately represent large deformations, as undergone by the IVD.

In 2021, Remus et al. [47] published a passive spine model created with ArtiSynth combining rigid vertebral bodies and deformable IVDs as FEM bodies. In Remus’ work, data were segmented from the VHM [67] and smoothed. Auxiliary vertebral bodies were additionally derived and acted as interfaces to the IVDs. Vertebral bodies and endplates were not differentiated. Facet joints were modeled in the shape of cylindrical rigid bodies. As FEM components, the IVDs were modeled with a Yeoh material model for the NP and a Mooney–Rivlin material model for the four lamellae of the AF, which was composed with multi-point springs linking the external nodes of the lamellae. Ligaments were modeled as multi-point springs or axial springs. The researchers validated their model in a quasi-static framework with multiple load cases and respective kinematics of in vitro literature data and numerical data. They calculated intradiscal pressure by using the negative mean of normal stresses of all FEM nodes in the NP. Values showed high alignment with in vitro literature data of IVDs at all vertebrae levels. Still, no muscular components were integrated into the model. In subsequent studies, the model was used to study the impact of degenerative changes of the IVD on the axis of rotation by altering its mechanical properties [75] and the effects of a simplified intra-abdominal pressure [76] with integrated muscles as active components. Both studies showed reasonable results, as the authors stated.

Refer to Table 2 for an overview of the reviewed bidirectional co-simulation studies. In summary, two main approaches have been used to couple MBS and FEM spine models bidirectionally: a gluing algorithm providing constant data exchange at certain RPs, managed by a co-simulation engine, and a particle-based approach dividing the model into rigid and flexible particles and solving the resulting ODE with a semi-implicit integrator.

## 5. Limitations and Challenges

Most co-simulation models of the spine use a unidirectional coupling approach, transferring data singularly from one simulation to the other by applying both simulation methods consecutively. MBS spine models can profit by defining joint stiffness parameters based on FEM simulations of the IVD. FEM models of the spine or spinal components can be improved when muscle or ligament forces and moments are implemented as BCs. However, when providing a time-dependent input, the time-dependency is influenced by the deformation properties of a material or component. Equal mechanics cannot be expected when comparing FEM and MBS IVD representations due to their different modeling approaches. The input accomplished by the results of the MBS initially carried out is therefore only partly suitable for being incorporated in a subsequent FEM simulation. This limitation is demonstrated in the effort that has been put into synchronizing the respective models of the unidirectional co-simulations. The manual adaption steps that become necessary may provoke inaccuracies and take much time. Kumaran et al. [64], for example, identified an issue with the interaction of forces and displacement in their simulation and stated that the muscle forces did not produce the desired motion of the vertebrae. They then implemented the simulation with a given displacement rather than a correct interaction of forces and displacement. The same synchronization difficulties were experienced by Khoddam-Khorasani et al. [40] and Liu et al. [41], who both mentioned the need for a manual, iterative process to achieve convergence between the FEM and MBS model or to account for factors such as compressive and shear stiffness. As in Liu et al. [41], the concept of a FL is frequently used to apply summarized loads in the direction of the spinal curvature to provide realistic loading conditions. However, it neither accounts for time-dependent changes in loads, nor dissolves the need of trial and error procedures [36]. In sum, unidirectional co-simulation is often associated with lower accuracy and convergence issues.

To overcome these limitations, a few recent studies have implemented bidirectional co-simulations of the spine. By updating the deformation values of the IVDs and the resulting positions of the vertebrae in every increment, updated reaction moments and forces of muscles, tendons and ligaments can be considered. Of the research groups using bidirectional co-simulation models, all authors found that their models were able to deliver accurate results [46,47,73]. However, two main methods were identified to bidirectionally couple the MBS and FEM solver. Monteiro et al. used an interface approach consisting of two linking RPs, at which the kinematic and kinetic data were exchanged constantly. Although this constant exchange of data solves many of the problems encountered in unidirectional co-simulation, a limiting factor could have been that only one reference node represented the interface between the vertebra and its endplate in this study. The pressure distribution on the IVD thus needed to be derived from one single node, which may have resulted in lowered accuracy. As already reported by Monteiro et al., the implementation of multiple RPs per linkage would be a reasonable adaption to account for more complex deformations.

The particle-based approach divides the whole model into two types of particles, rigid and flexible ones [47,73,74]. Thus, the interface between the flexible FEM and the rigid MBS bodies consists of more than one RP. A mapping step combines the distinct particle sets into one model consisting of a single ODE, which is solved by an integrator. A limiting factor in this approach is the use of a lumped mass matrix and linear co-rotation definitions (neglecting other rotation effects) for the FEM component, which has been associated with less adequate representation of large deformations, as they appear in the IVD. [77]

Despite these limitations, bidirectional co-simulation models of the spine can provide a holistic understanding of the spine because they consider both the overall kinematics with the muscular and gravitational forces and moments, and the detailed mechanics of the IVDs with their deformations and stress distributions.

## 6. Conclusions and Future Directions

FEM is broadly used in detailed analyses of internal stresses in deforming bodies but lacks computational efficiency. MBS is more efficient, but can only achieve a certain degree of detail when it comes to deformations of model components. A combination of both methods, not only in a unidirectional way, but in a bidirectional manner of data exchange, can provide both accuracy and efficiency.

Future studies will likely include widespread use of bidirectional co-simulation models to understand and predict the behavior of the spine. Including automated segmentation algorithms such as the one implemented by Sekuboyina et al. [26] could accelerate two things: individual, more adequate diagnoses due to patient-specific geometries, and clinical investigations containing large cohorts. Those detailed, personalized simulations of large cohorts could be used to better understand the underlying mechanisms of pathological changes and the biomechanics of overload situations in ambitious athletes, or to predict injuries before they occur.

## Figures and Tables

**Figure 1 bioengineering-10-00315-f001:**
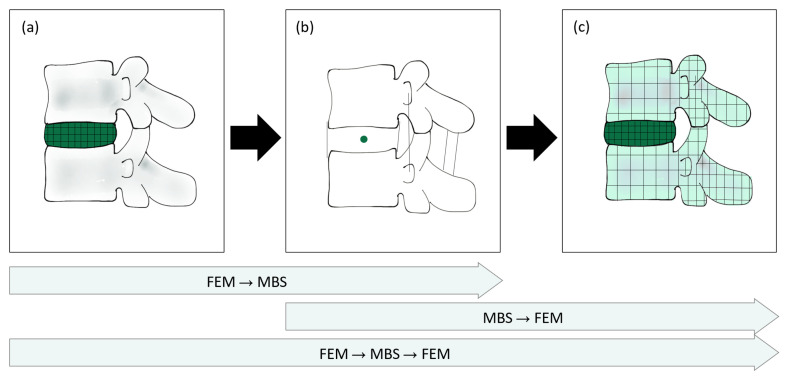
Schematic overview of unidirectional coupling methods. Arrows on the bottom indicate the respective coupling methods. (**a**) Representative FEM model of a FSU containing a detailed IVD, as found in Karajan et al. [44]. Resulting FEM displacements were converted to MBS bushing element parameters. (**b**) Typical MBS model with a joint representing the IVD. Results from the simulation, such as muscle forces or time-dependent displacements of vertebrae or joints, were subsequently included in an FEM simulation. (**c**) Representative FEM model of a FSU or a larger section of the spine. With the BCs defined according to the results of (**b**), these models were often used to calculate IDPs, stress distributions in the IVDs and load-sharing mechanisms.

**Figure 2 bioengineering-10-00315-f002:**
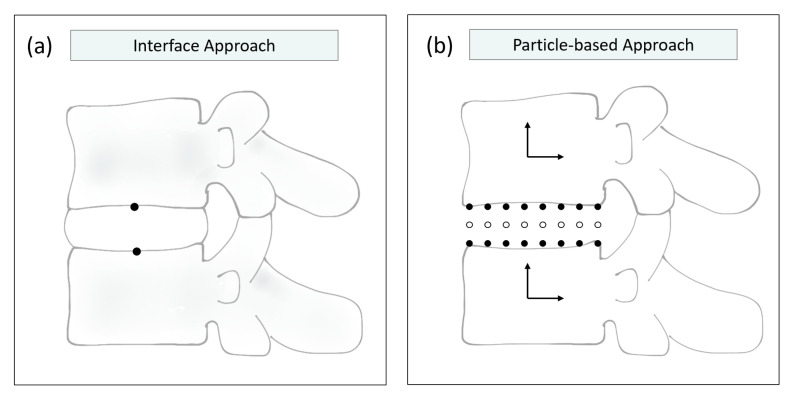
Graphical representation of bidirectional coupling methods found in the literature. (**a**) Coupling algorithm implemented by Monteiro et al. [46]. Two RPs, pictured as black dots, served as an interface between the flexible IVD and the rigid vertebrae. (**b**) Coupling method realized by Dicko et al. [73], which was inspired by an algorithm of Stavness et al. [74]. The model was divided into two types of particles. Rigid particles are illustrated as black dots and flexible particles as circles.

**Table 1 bioengineering-10-00315-t001:** Recent simulation studies using unidirectional co-simulation to investigate the spine with information on the execution order, transferred data, the software structure and the source of the model geometry.

	MBS Solver ^1^	FEM Solver	Execution Order	Transferred Data	Software Structure	Model Geometry
Esat et al., 2005, 2009 [37,38]	visualNastran 4D from MSC Software	Marc/Mentat from MSC Software.	MBS → FEM	Two time-dependent sagittal forces and one sagittal moment at each IVD → BC	Distinct software, manual transfer	Literature
Du et al., 2014 [49]	Hypermesh	Hypermesh (Altair Engineering)	MBS → FEM	Time-dependent translation & rotation at hip joint & T9 endplate → BC	Distinct software, manual transfer	Literature
Henao et al., 2016 [51]	ADAMS [52]	RADIOSSTM (Altair Engineering)	MBS → FEM	Displacement of vertebrae	Distinct software, manual transfer	Patient-specific/Literature
Honegger et al., 2021 [53]	OpenSim	Abaqus	MBS → FEM	Time-dependent joint angles and muscle forces	Distinct software, manual transfer	Preexisting FEM model fitted to patient-specific geometry
Kamal et al., 2019 [39]	Matlab	Abaqus	MBS → FEM	Resulting muscle forces and reaction moments as distributed pressure and shear stress	Distinct software, manual transfer	CT-based
Azari et al., 2018, Khoddam-Khorasani et al., 2018, Rajaee et al., 2021 [36,40,63]	Abaqus/Matlab	Abaqus/In-house	MBS → FEM	Mostly static muscle forces and moments	Distinct software/One software incorporating muscles and detailed passive elements	CT-based
Karajan et al., 2013 [44]	Not mentioned	Not mentioned	FEM → MBS	IVD displacement → bushing component definition	Distinct software, manual transfer	Simplified as cylinders
Kumaran et al., 2021 [64]	OpenSim	Abaqus	FEM → MBS → FEM	FEM → MBS: ROM MBS → FEM: Muscle forces	Distinct software, manual transfer	Literature
Liu et al., 2018,2020 [41,42]	Anybody	Abaqus/Hypermesh	FEM → MBS → FEM	FEM → MBS: Joint stiffness curves of IVDs MBS → FEM: Reaction moment, ligament and muscle forces at T12-L1 joint	Distinct software, manual transfer by trial-and-error	˙Default Anybody data/Literature
Meszaros et al., 2021 [43]	VHM	Abaqus	FEM → MBS → FEM	FEM → MBS: IVD response as mechanical parameters MBS → FEM: Time-dependent muscle, tendon & ligament forces	Distinct software, manual transfer	VHM for MBS, patient-specific & VHM-based FEM (morphed)

^1^ In this definition, we also include models with clear characteristics of MBS models, such as the involvement of
mainly rigid bodies and their interconnections by joint-like components.

**Table 2 bioengineering-10-00315-t002:** Simulation studies using bidirectional co-simulation to investigate the spine with information on the execution order, transferred data, the software structure and the source of the model geometry.

	MBS Solver	FEM Solver	Execution Order	Transferred Data	Software Structure	Model Geometry
Monteiro et al., 2011 [46]	Abaqus	Apollo (Fortran)	constant	Displacements in MBS ↔ reaction forces and moments in FEM	Single software	Literature
Dicko et al., 2015 [73]	Not mentioned	Not mentioned	constant	Integrated approach based on particles	Single software	Literature
Remus et al., 2021 [47]	ArtiSynth	ArtiSynth	constant	Integrated approach based on particles	Single software	Literature (VHM)

## Data Availability

Not applicable.

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
