# Peer review of "Recent Advances in Coupled MBS and FEM Models of the Spine—A Review"

_bioengineering, 2023, doi:10.3390/bioengineering10030315_

Round 1
Reviewer 1 Report
The authors are invited to describe the method used to find relevant papers including the sources, databases, keywords, time range, and selection criteria.
Studies using Artificial Neural Network were not included. The review paper summarizes the methodology and results of some studies and stated the discussion and limitation reported by the authors of those papers. However, in a review article, it is expected that the authors comment and provide their own opinion on these studies.
There are few typos, spelling and grammatical mistakes. Also, please check the following:
L443 and table 2 : VHP instead of VHM
Reviewer 2 Report
This paper intruduces recent studies about coupled MBS and FEM models of the spine, it can be accepted if the following questions are addressed well.
1. Could you enrich the contents in the part of "FEM → MBS";
2. For better readability of this paper, could you add some figures to present the relevant representative studies?
3. Simple MBS models often lose computational accuracy, but accurate FEM models tend to reduce the computational efficiency, for different analysis cases how to choose the appropriate model can guarantee both efficiency and accuracy?
Reviewer 3 Report
The authors conducted a detailed analysis of the MBS and FEM and their combinations to understand spinal disorders and the interaction of various factors, such as motion or posture, back pain and the biomechanics of the intervertebral discs. Review seemed to be positive in the scientific fields. However, the following issues were detected.
1. In the review, there is little modern literature, while a significant part of the literature is 20-30 years old, including references to works from 1985-1998.
2. The review describes in detail the results of studies by various authors and practically lacks an analysis of these results, with the exception of a small conclusion.
3. To improve the perception of the material, it is advisable to present illustrative material.
Reviewer 4 Report
The topic is very specific and difficult to me that I'm not an expert in this engeneering field.
That said this is a narrative review, but the format is atypical and very difficult to read of MD. I suggest the author to comply to guidelines for writing reviews and to organize the paper with a method section in which you define the search method applied and how you decided to summarize the information, then present results and discuss the results comparing the different published findings.
Check this website for guidance https://www.prisma-statement.org/?AspxAutoDetectCookieSupport=1
Round 2
Reviewer 1 Report
The authors have addressed my concerns. I have no additional comments.